# TGF-β expressed by M2 macrophages promotes wound healing by inhibiting TSG-6 expression by mesenchymal stem cells

Young Woo Eom[1], Ju-Eun Hong[2], Pil Young Jung[3], Yongdae Yoon[1], Sang-Hyeon Yoo[2], Jiyun Hong[2], Ki-Jong Rhee[2], Bhupendra Regmi[1], Saher Fatima[1], Moon Young Kim[1,4], Soon Koo Baik[1,4], Hoon Ryu[3], Hye Youn Kwon[3]*

1 Regeneration Medicine Research Center, Yonsei University Wonju College of Medicine, Wonju, Korea, 2 Department of Biomedical Laboratory Science, Yonsei University Mirae Campus College of Health Sciences, Wonju, Korea, 3 Department of Surgery, Yonsei University Wonju College of Medicine, Wonju, Korea, 4 Department of Internal Medicine, Yonsei University Wonju College of Medicine, Wonju, Korea

☙ These authors contributed equally to this work.
* kwonhy@yonsei.ac.kr (HYK)

## Abstract

Wound healing involves the collaboration of multiple cells, including macrophages and fibroblasts, and requires the coordination of cytokines, growth factors, and matrix proteins to regulate the repair response. In this study, we investigated how M2 macrophages regulate expression of the anti-fibrotic and anti-inflammatory regulator tumor necrosis factor-α (TNF-α)-stimulated gene 6 (TSG-6) secreted by adipose tissue-derived stem cells (ASCs) during wound healing. Interleukin (IL)-4/IL-13, which is used to differentiate macrophage M2 phenotypes, increases TSG-6 in ASCs; however, M2 macrophages significantly decrease TSG-6 in ASCs. Transforming growth factor (TGF)-β expression was increased, and TNF-α expression was decreased in M2 macrophages. TGF-β inhibited IL-4/IL-13-induced ASC TSG-6 expression. In addition, TSG-6 suppressed TGF-β-triggered wound closure and fibrogenic responses in LX-2 cells. Collectively, TSG-6 inhibited wound healing, but M2 macrophage-expressed TGF-β prevented TSG-6 production from ASCs, which ultimately helped wound healing. Our results indicate that the balance of TNF-α and TGF-β levels during wound healing regulates TSG-6 production from ASCs, which may ultimately modulate the healing process. Our study findings could contribute to novel therapeutic strategies that manipulate the delicate balance between TNF-α and TGF-β to enhance wound repair and mitigate fibrosis.

## Introduction

Although the cells and secreted factors involved in wound healing are slightly different depending on the cause of the wound (trauma, burn, or pathogen) and organ injured (skin, lungs, liver, or brain), three distinct phases of healing occur in all wound types: coagulation and inflammation, proliferation, and remodeling [1–4]. As macrophages play a role in wound healing by clearing pathogens, scavenging dead cells, and initiating and maintaining tissue remodeling and regeneration, wound healing is impaired when macrophages

**Data availability statement:** All relevant data are within the manuscript and its Supporting Information files.

**Funding:** This work was supported by the Basic Science Research Program through the National Research Foundation of Korea (NRF) (RS-2023-00250982 and 2021R1I1A1A01056265).

**Competing interests:** The authors have declared that no competing interests exist.

are depleted [5–7]. Macrophages are of two types, pro-inflammatory (M1) macrophages [8,9] and anti-inflammatory (M2) macrophages [10]. Upon injury, M1 macrophages are recruited to the wound, where they infiltrate the wound area; clear pathogens, foreign debris, and dead cells; release inflammatory cytokines (e.g., interleukin (IL)-1β, IL-6, IL-12, and tumor necrosis factor (TNF)-α); and induce a Th1-type response [8,9]. Subsequently, when tissue regeneration is initiated, macrophages convert to M2 macrophages, which promote fibroblast proliferation, angiogenesis, extracellular matrix (ECM) repair, anti-inflammatory cytokine release (TGF-β, IL-10, IL-1 receptor antagonist [IL-1ra], and the IL-1 type II decoy receptor), and inflammation resolution [10]. However, chronic wounds do not heal as easily because they remain in an early inflammatory state [11,12]. Therefore, therapies that convert M1 macrophages into M2 macrophages may prove effective in chronic wound healing.

Mesenchymal stem cells (MSCs) secrete various trophic factors, leading to tissue regeneration and alleviating inflammatory responses, and they have been utilized in the treatment of various diseases [13,14]. MSCs migrate to the wound area where they inhibit inflammation by secreting variety molecules, such as human leukocyte antigen G, IL-1ra, IL-6, IL-10, indoleamine 2,3-dioxygenase (IDO), nitric oxide (NO), prostaglandin E2 (PGE2), and TNF-α stimulated gene-6 (TSG-6) [15–17], thereby leading to regeneration [13,14].

TSG-6, which is expressed by monocytes, neutrophils, macrophages, fibroblasts, and MSCs, has tissue-protective and anti-inflammatory properties [18–21]. Importantly, TSG-6 modulates the migration and proliferation of endothelial cells, neutrophils, mast cells, vascular smooth muscle cells, and macrophages, and its anti-inflammatory effects seem to have a role in this modulation [22,23]. To regulate fibrosis, TSG-6 interacts with diverse components of the ECM to either stabilize and/or remodel the ECM structure [24,25]; decreases the expression of TGF-β1 (a pro-fibrotic molecule), while increasing that of TGF-β3 (an anti-fibrotic molecule) [26]; and reduces myofibroblast differentiation and collagen deposition [26]. Additionally, TSG-6 can effectively inhibit the proliferation of hypertrophic scar fibroblasts through modulation of IRE1α/TRAF2/NF-κB signaling [27] and regulate the migration of neutrophils and vascular smooth muscle cells [28,29]. Although TGF-β1 regulates fibroblast migration and wound healing [30], there are no reports that TSG-6 directly regulates fibroblast migration. However, it is predicted that TGF-β1-induced fibroblast migration is regulated by TSG-6 because TSG-6 can inhibit SMAD phosphorylation [21] and regulate the TGF-β1/TGF-β3 ratio [26]. TSG-6-deficient mice exhibit abnormal inflammatory responses and delayed wound closure [31], suggesting the crucial role of TSG-6 in regulating inflammation and ECM during wound healing.

TSG-6 expression in MSCs is upregulated by interferon (IFN)-γ, IL-1β, LPS, and TNF-α, but is suppressed by TGF-β [32]. TSG-6 induces M2 macrophage polarization, reduces IL-1β expression [32], and inhibits fibroblast ECM production [21]. Importantly, co-culturing M1 macrophages with MSCs significantly increases TSG-6 expression in both of these cell types. Despite the inherent inflammatory nature of M1 macrophages, this increase in TSG-6 expression reduces fibrosis in LX-2 hepatic stellate cells [21], suggesting that TSG-6 regulates fibrosis by modulating inflammation during wound healing.

Preliminary finding show that IL-4/IL-13, which induces the M2 macrophage polarization, increases TSG-6 production in MSCs, suggesting that TSG-6 may delay wound healing during the proliferation and remodeling phases. Consequently, since TSG-6 inhibits the fibrotic activity of LX-2 cells, its expression in MSCs during these phases may delay wound healing. Therefore, this study explores the roles of TSG-6 on proliferation and remodeling in wound healing, by investigating whether TGF-β-expressing M2 macrophages regulate TSG-6 expression in MSCs, whether TSG-6 regulates LX-2 cell proliferation, fibrosis, and migration, and

whether conditioned media from co-cultures of MSCs and M2 macrophages modulate LX-2 cell activities on proliferation and migration.

## Materials and methods

### Reagents

The reagents used in this study and their suppliers are as follows: IFN-γ, TNF-α, TGF-β, and TSG-6 from R&D Systems (Minneapolis, MN, USA); antibodies against GAPDH (sc47724) and TSG-6 (sc377277) from Santa Cruz Biotechnology (Santa Cruz, CA, USA); antibodies against SMAD2/3 (8685S) and COL1A1 (39952S) from Cell Signaling Technology (Danvers, MA, USA); antibodies against fibronectin (F6140) from Sigma-Aldrich (St. Louis, MO, USA); and antibodies against α-smooth muscle actin (α-SMA, ab5694) from Abcam (Cambridge, UK); antibodies against pSmad2/3 (PA5–110155) from Thermo Fisher Scientific (Waltham, MA, USA). All other materials were purchased from Sigma-Aldrich.

### Cell culture

Studies commenced following ethical approval from "Institutional Review Board of Yonsei University Wonju College of Medicine" (CR321076). Human adipose tissues sample were obtained from three healthy donors, ages 24–38 years, who underwent elective liposuction procedures at the Wonju Severance Christian Hospital (Wonju, Korea) between October 1, 2021 and December 31, 2023. Written consent was obtained before the surgical procedure. A modified protocol, as described by Zuk et al. [33] was used to isolate adipose tissue-derived mesenchymal stem cells (ASCs) and subcultured in low-glucose Dulbecco's Modified Eagle's Medium (DMEM; Gibco BRL, Rockville, MD, USA) containing 10% fetal bovine serum (FBS; Gibco BRL) and penicillin/streptomycin. ASCs at less than five passages were used for this experiment.

THP-1 monocytes were purchased from the Korean Cell Line Bank (Seoul, Republic of Korea) and maintained in Roswell Park Memorial Institute 1640 Medium (RPMI 1640; Gibco BRL) supplemented with 10% FBS and penicillin/streptomycin (Gibco BRL). THP-1 cells were treated with 100 nM phorbol 12-myristate-13-acetate (PMA) for two days for differentiation into macrophages and then treated with IFN-γ (20 ng/mL) and LPS (10 pg/mL) or IL-4 (20 ng/mL) and IL-13 (20 ng/mL) for additional two days to transition from PMA-treated THP-1 cells into M1 or M2 macrophages. For indirect co-culture of ASCs and macrophages, ASCs and macrophages were cultured in different plates, i.e., ASCs were seeded in 6-well plates (SPL Life Sciences, Pocheon, Korea) in DMEM one day prior to indirect co-culture, and THP-1 monocytes cultured in RPMI 1640 were differentiated into macrophages using PMA in the upper chamber of transwell plates (4.5 μm pore size) two days prior to indirect co-culture (the lower chamber was cell-free). For indirect co-culture, the upper chamber with macrophages was mounted in 6-well plates seeded with ASCs and immediately treated with IFN-γ/LPS or IL-4/IL-13 to differentiate macrophages into M1 or M2 state. For co-cultures, DMEM and RPMI 1640 were diluted 1:1.

The human hepatic stellate cell line LX-2 was purchased from Millipore (Burlington, MA, USA). LX-2 cells were maintained in high-glucose DMEM (Gibco BRL) supplemented with 3% FBS (Gibco BRL) and penicillin/streptomycin (Gibco BRL) at 37°C and in a 5% CO2 incubator.

Conditioned media (CM) was prepared by mono- or co-culture of ASCs and macrophages, using 6-well plates or transwell plates (SPL, Pocheon, Korea), respectively. The CM was collected, filtered with syringe-driven filters (0.45 μM) after two days, and stored at -80°C for further use. CM (500 μL) was concentrated to approximately 15 μL using an Amicon Ultra

– 3K (Millipore) before treating CM into wound healing assay and then added in one well of a 24 well plate with 500 μL of medium.

## Immunoblotting

Cells were lysed in sodium dodecyl sulfate-polyacrylamide gel electrophoresis (SDS-PAGE) sample buffer (62.5 mM Tris-HCl, pH 6.8; 1% SDS; 10% glycerol; and 5% β-mercaptoethanol) for protein extraction. Protein samples were boiled for 5 min, subjected to SDS-PAGE, and transferred to an Immobilon polyvinylidene difluoride membrane (PVDF, Millipore). Skim milk (5%) in Tris-HCl buffered saline containing 0.05% Tween 20 was used to block the membrane and later incubated with primary antibodies against TSG-6 and GAPDH (1:1000, Santa Cruz Biotechnology, Dallas, TX, USA); α-SMA (1:2000, Abcam), COL1A1 and SMAD2/3 (1:1000, Cell Signaling Technology); fibronectin (1:1000, Sigma-Aldrich); and pSmad2/3 (Thermo Fisher Scientific), followed by incubation with peroxidase-conjugated secondary antibodies (1:2000, Cell Signaling Technology). EZ-Western Lumi Pico or Femto (DOGEN, Seoul, Korea) was used to treat the membrane and was visualized using a ChemiDoc XRS+ system (Bio-Rad, Hercules, CA, USA).

## Quantitative PCR

TRIzol reagent (Gibco BRL) was used to extract total RNA according to the manufacturer's instructions. Total RNA (1 μg) was required for cDNA synthesis using the Verso cDNA synthesis kit (Thermo Fisher Scientific, Waltham, MA, USA). The sequences of sense and anti-sense primers were as follows: 5'-AAGTGGACATCAACGGGTTC-3' and 5'-GTCCAG GCTCCAAATGTAGG-3' for TGF-β1 (NM_000660.7), 5'-CCCATGTTGTAGCAAACCCT-3' and 5'-TGAGGTACAGGCCCTCTGAT-3' for TNF-α (NM_000594.4), and 5'-CAAG GCTGAGAACGGGAAGC-3' and 5'-AGGGGGCAGAGATGATGACC-3' for GAPDH (NM_001256799.3). The reaction mixture (10 μL) included cDNA, primer pairs, and SYBR Green PCR Master Mix (Applied Biosystems, Dublin, Ireland). A QuantStudio 6 Flex Real-time PCR System (Thermo Fisher Scientific) was used to conduct polymerase chain reaction (PCR). All qPCR reactions were performed in triplicates. GAPDH expression was used for normalization. The $2^{-(\Delta\Delta Cq)}$ method was used to calculate the relative fold changes in mRNA expression.

## Methylthiazolyldiphenyl tetrazolium bromide (MTT) assay

LX-2 cells ($8 \times 10^3$ cells/cm$^2$) were seeded in 96-well plates and incubated for 24 h, before treatment with TGF-β (1 ng/mL) and/or TSG-6 (40 ng/mL) for an additional 24 h. MTT (5 mg/mL in PBS) was added to each well, and the cells were further incubated at 37°C for 2 h. After incubation for 15 min, MTT formazan crystals were dissolved in 100 μL of dimethyl sulfoxide. The optical density of each well was then measured at 570 nm using a microplate reader (Molecular Devices, San Jose, CA, USA).

## Scratch wound closure assay

LX-2 cells were seeded ($4 \times 10^5$ cells/cm$^2$) into 12-well plates and cultured until a confluent monolayer formed. The confluent LX-2 monolayers were scratched manually using a universal sterile 10 μL pipette tip and then rinsed with PBS. Cells were treated with TGF-β and/or TSG-6 and conditioned media obtained from the culture of macrophages and/or ASCs. Cell migration was observed under an inverted microscope (Eclipse TS2R, Nikon, Tokyo, Japan) at 0, 12, and 24 h post-wound formation. Wound images taken with a digital camera (DS-Ri2, Nikon) were analyzed using ImageJ software (National Institutes of Health, https://imagej.net/ij/index.html, 10 July 2024) to measure the area of the scratched field.

## Statistical analysis

No predetermined sample size calculation was done. All experiments were performed three or four times, and representative results were presented. Data are expressed as the mean ± standard deviation (SD) or standard error (SE) of the mean. To compare the group means, Student's t-test and one-way analysis of variance were performed, followed by Scheffe's test. $P < 0.05$ was considered statistically significant.

## Results

### TSG-6 expression in ASCs co-cultured with macrophages

Previously, we observed that ASCs co-cultured with M1 macrophages can significantly express TSG-6 [21]. As TSG-6 is capable attenuating fibrosis, we investigated whether M2 macrophages, which play an essential role in fibrosis and wound healing, could regulate TSG-6 expression in ASCs. M2 macrophages differentiated with IL-4 and IL-13 expressed CD163 (60.53 ± 2.35%) and CD206 (54.87 ± 1.82%) (S1 Fig). First, we analyzed TSG-6 mRNA expression in ASCs under conditions conducive to M1 and M2 macrophage differentiation and in ASCs co-cultured with macrophages in the M1 or M2 state. In ASCs, both M1- (IFN-γ/LPS) and M2-differentiation factors (IL-4/IL-13) increased TSG-6 mRNA expression, with M2-differentiation factors increasing TSG-6 mRNA expression more than M1-differentiation factors. However, TSG-6 mRNA expression was increased by M1 macrophages more significantly in co-cultured ASCs than in mono-cultured ASCs, whereas M2 macrophages decreased TSG-6 mRNA expression in co-cultured ASCs (Fig 1A). While IFN-γ and LPS, which convert macrophages to the M1 phenotype, induced TSG-6 expression to a negligible extent in mono-cultured ASCs (lane 2 in Fig 1B), M1 macro-phages markedly increased the expression of TSG-6 in ASCs (lane 5 in Fig 1B). In contrast, IL-4 and IL-13, which drives the polarization

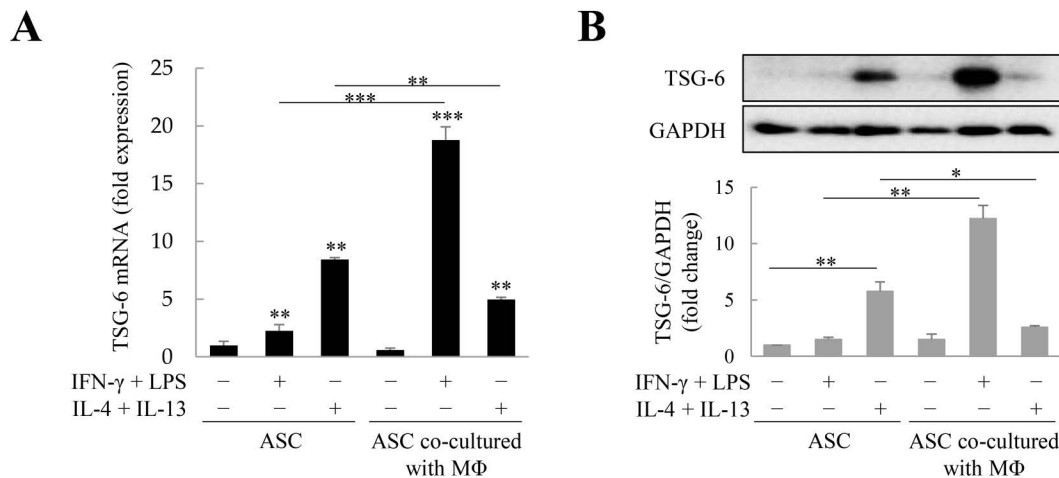

**Fig 1. Tumor necrosis factor- α (TNF-α)-stimulated gene 6 (TSG-6) expression in adipose tis-sue-derived stem cells (ASCs) co-cultured with macrophages.** ASCs were treated with interferon (IFN)-γ/lipopolysaccharide (LPS) or interleukin (IL)-4/IL-13 or co-cultured with macrophages for two days. TSG-6 mRNA (A) and protein (B) expression in ASCs under treatment with IFN-γ/LPS or IL-4/IL-13 and/or co-culture with macrophages. In panels A and B, IFN-γ + LPS treatment in the 5th sample induced macrophages to differentiate to M1 macrophages and IL-4 + IL-13 treatment in the 6th sample induced macrophages to differentiate to M2 macrophages. Immunoblotting was used to analyze TSG-6 expression in ASCs. Relative expression was normalized with respect to glyceraldehyde 3-phosphate dehydrogenase (GAPDH) expression. Data are presented as the mean ± standard deviation (SD) of three independent experiments. *$p \leq 0.05$, **$p \leq 0.01$, and ***$p \leq 0.001$. MΦ, macrophages (PMA-treated THP-1 cells).

of macrophages to switch from the M1 to M2 phenotype, induced TSG-6 expression on mono-cultured ASCs (lane 3 in Fig 1B), but M2 macrophages barely induced TSG-6 production from ASCs (lane 6 in Fig 1B). These findings indicate that despite IL-4/IL-13 induces TSG-6 production in ASCs, whereas M2 macrophages inhibit TSG-6 production.

## TSG-6 expression inhibition via TGF-β in IL-4 and IL-13-treated ASCs

We previously showed that pro-inflammatory factors (IFN-γ, IL-1β, LPS, and TNF-α) markedly induced TSG-6 expression in ASCs, but TGF-β inhibited it [32]. Therefore, we examined whether TGF-β could also inhibit IL-4/IL-13-induced TSG-6 production in ASCs. We first analyzed TGF-β and TNF-α expression in M1 and M2 macrophages. In M1 macrophages differentiated with IFN-γ/LPS, TGF-β expression decreased (Fig 2A), but TNF-α expression increased (Fig 2B). In contrast, in IL-4/IL-13-differentiated M2 macrophages, TGF-β expression increased (Fig 2A), but TNF-α expression decreased (Fig 2B). In M1 macrophages co-cultured with ASCs, TNF-α expression increased more than that in mono-cultured M1 macrophages (Fig 2B). However, TGF-β expression in M2 macrophages co-cultured with ASCs was lower than that in M2 macrophages cultured alone, but remained above the level expressed in macrophages (Fig 2A). These results are in accordance with the findings of a previous study, that is, M1 macrophages increase pro-inflammatory TNF-α expression and M2 macrophages increase anti-inflammatory TGF-β expression [34]. Next, we determined whether IL-4/IL-13-induced TSG-6 expression in ASCs is regulated by TGF-β and TNF-α. As expected, TNF-α synergistically increased TSG-6 expression induced by IL-4/IL-13 in ASCs, whereas TGF-β de-creased TSG-6 expression (Fig 2C). At concentrations > 0.4 ng/mL, TGF-β inhibited IL-4/IL-13-induced TSG-6 expression in ASCs (Fig 2D). Our results indicate that TGF-β-expressing M2 macrophages inhibit TSG-6 expression by ASCs.

## Effects of TSG-6 on fibrosis, proliferation, and migration of LX-2 cells

In wound healing, TGF-β regulates various functions, such as cell proliferation, cell differentiation, ECM production, and immune response modulation [21,32,35]. We investigated whether TSG-6 can regulate TGF-β-induced fibrosis, proliferation, and migration of LX-2 cells. In LX-2 cells, TGF-β upregulated phosphorylation of SMAD2/3 and expression of α-SMA, fibronectin, and COL1A1, whereas TSG-6 markedly decreased these effects (Fig 3A). Similarly, TGF-β induced proliferation of LX-2 cells, but TSG-6 significantly decreased this proliferation (Fig 3B). Additionally, in the scratch wound healing assay, TGF-β increased LX-2 cell migration to induce wound closure, but TSG-6 suppressed their migration induced by TGF-β (Fig 3B). These results indicate that TSG-6 may delay TGF-β-induced wound healing, especially during the proliferation and remodeling phases of wound healing.

## Effects of ASCs treated with IL-4/IL-13 or co-cultured with M2 macrophages on LX-2 cell migration

Since TSG-6 inhibited LX-2 cell migration and IL-4/IL-13 increased TSG-6 production in ASCs, we analyzed whether CMs from ASCs treated with IL-4/IL-13 or co-cultured with M2 macrophages inhibited LX-2 cell migration. Exogenously added TSG-6 had no effect on LX-2 cell migration, whereas the CM of mono-cultured ASCs or M2 macrophages promoted the migration rate, and the CM of ASCs co-cultured with M2 macrophages also promoted the migration rate of LX-2 cells. However, the migration of LX-2 cells exposed to the CM of IL-4/IL-13-treated ASCs, in which TSG-6 expression increased (Fig 4), impairing migration rate. These results suggest that TSG-6 inhibits fibroblast fibrosis and migration during the wound healing process; therefore, the absence of TSG-6 is necessary for normal wound healing

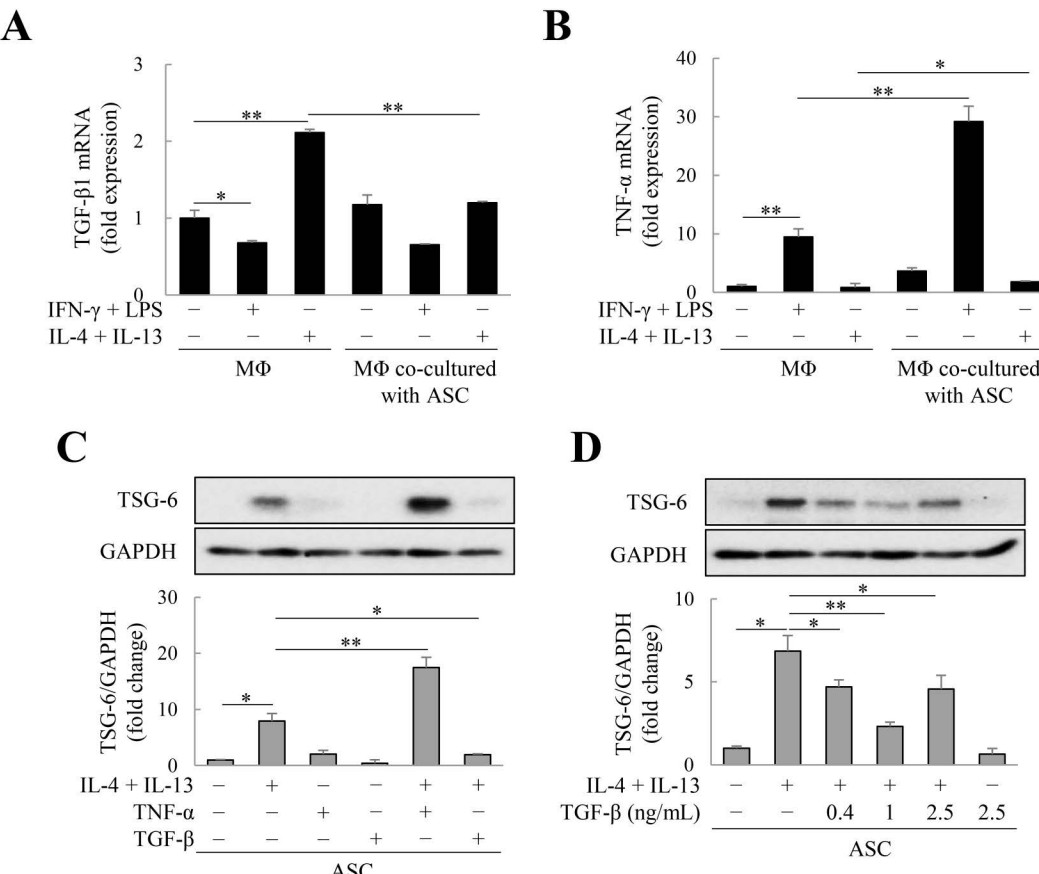

**Fig 2. Inhibition of tumor necrosis factor- α (TNF-α)-stimulated gene 6 (TSG-6) expression via transforming growth factor (TGF)-β in adipose tissue-derived stem cells (ASCs) treated with interleukin (IL)-4 and IL-13.** Macrophages and ASCs were seeded in the lower and upper chamber (transwell insert) of 6-well plate, respectively, and the TGF-β1 and tumor necrosis factor (TNF)-α expression in macrophages was analyzed using quantitative polymerase chain reaction (PCR). (A) TGF-β or (B) TNF-α mRNA in macrophages treated with IFN-γ/LPS or IL-4/IL-13 or co-cultured with ASCs. Relative expression was normalized with respect to glyceraldehyde 3-phosphate dehydrogenase (GAPDH) expression. Data are presented as the mean ± standard deviation (SD) of four independent experiments. *$p \leq 0.05$ and **$p \leq 0.01$. To observe TNF-α and TGF-β-induced alterations of TSG-6 production from ASCs, ASCs were treated with 1 ng/mL of TGF-β or 10 ng/mL of TNF-α for two days. (C) TSG-6 expression in ASCs treated with TNF-α or TGF-β. (D) TSG-6 expression in ASCs treated with TGF-β (0.4–2.5 ng/mL). Immunoblotting was used to analyze TSG-6 expression in ASCs. Relative expression was normalized with respect to GAPDH expression. Data are presented as the mean ± SD of three independent experiments. *$p \leq 0.05$ and **$p \leq 0.01$. MΦ, macrophages (PMA-treated THP-1 cells).

progression, and TGF-β regulates TSG-6 expression via endogenous or transplanted MSCs (Fig 5).

## Discussion

M1 macrophages upregulated TSG-6 production from ASCs, whereas M2 macrophages inhibited it. TSG-6 inhibited α-SMA, fibronectin, and COL1A1 expression in hepatic stellate LX-2 cells and inhibited their TGF-β-induced migration as well, delaying wound closure. These results indicate that TSG-6 may delay normal wound healing by regulating ECM component expression and inhibiting fibroblast migration.

Our previous studies showed that TSG-6 attenuates the inflammatory response of macrophages and promotes the M1-to-M2 transition [21,32]. We also observed that TSG-6 exerts

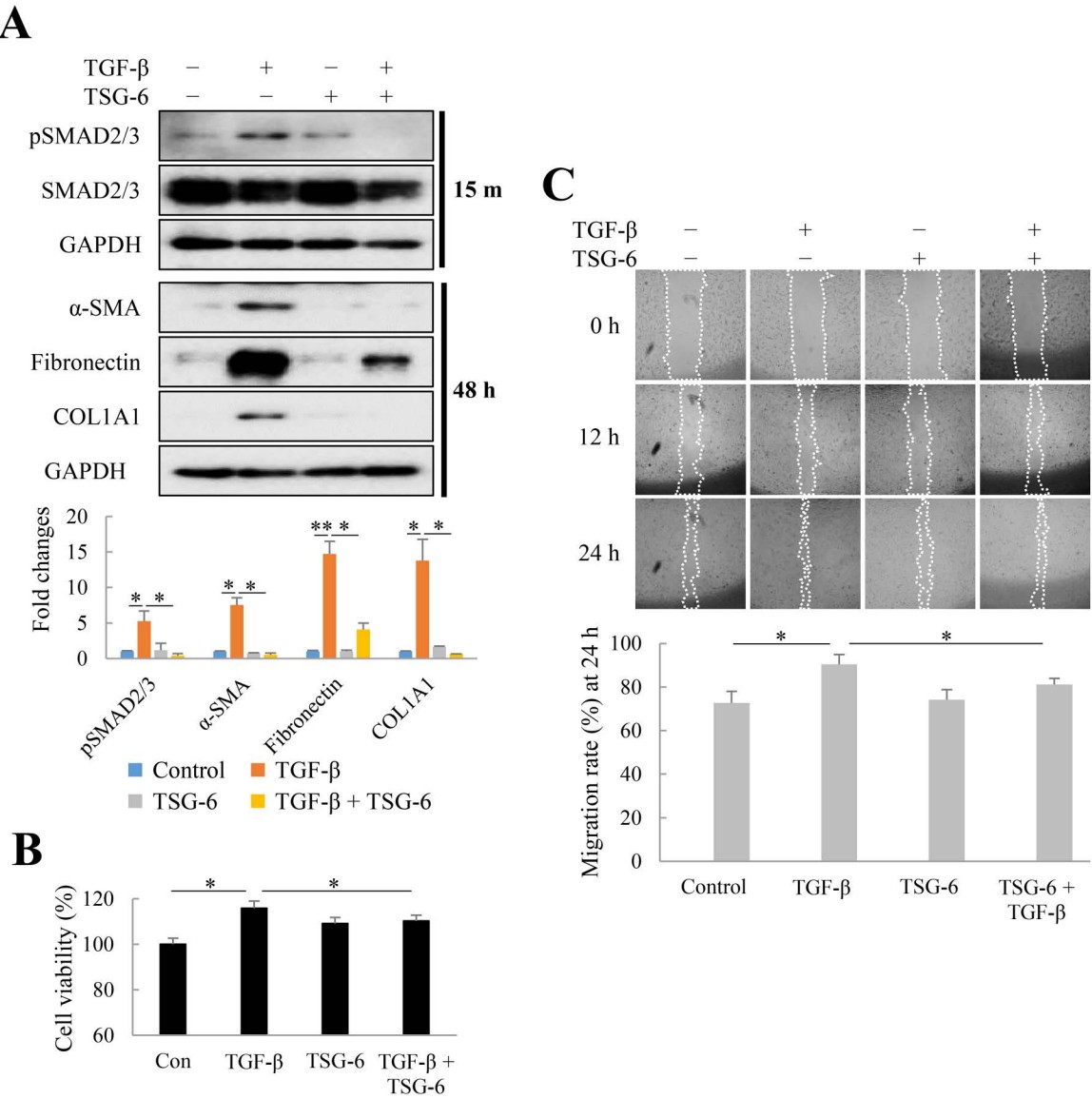

**Fig 3. Effects of tumor necrosis factor- α (TNF-α)-stimulated gene 6 (TSG-6) on the fibrosis, proliferation, and migration of LX-2 cells.** To determine the anti-fibrotic effect of TSG-6 and its effect on fibrosis, proliferation, and migration, LX-2 cells were treated with transforming growth factor (TGF)-β (1 ng/mL) and/or TSG-6 (40 ng/mL). (A) Anti-fibrotic effects of TSG-6 in LX-2 cells. (B) Effects of TSG-6 on LX-2 cell proliferation after 48 h. Data are presented as the mean ± SD of four replicates. (C) Effects of TSG-6 on LX-2 cell migration. Representative images are shown from three independent experiments and the cell-free region (outlined wound area) was analyzed using ImageJ. Values of percentage wound closure were shown as the mean ± standard deviation (SD). *$p \leq 0.05$. and **$p \leq 0.01$.

anti-fibrotic activity through the inhibition of SAMD3 phosphorylation in LX-2 cells [21]. In the context of wound healing, the attenuation of the inflammatory reactions is a key therapeutic strategy. TSG-6, expressed by MSCs, can mitigate the inflammatory response at the wound site by decreasing macrophage inflammation and facilitating the M1-to-M2 transition [36]. Subsequently, MSCs promote the proliferation of various cell types, including fibroblasts, and induce ECM remodeling [37]. However, given that TSG-6 also inhibits fibroblast proliferation [27] and its expression is suppressed by M2 macrophages [32], TSG-6 is expected to regulate

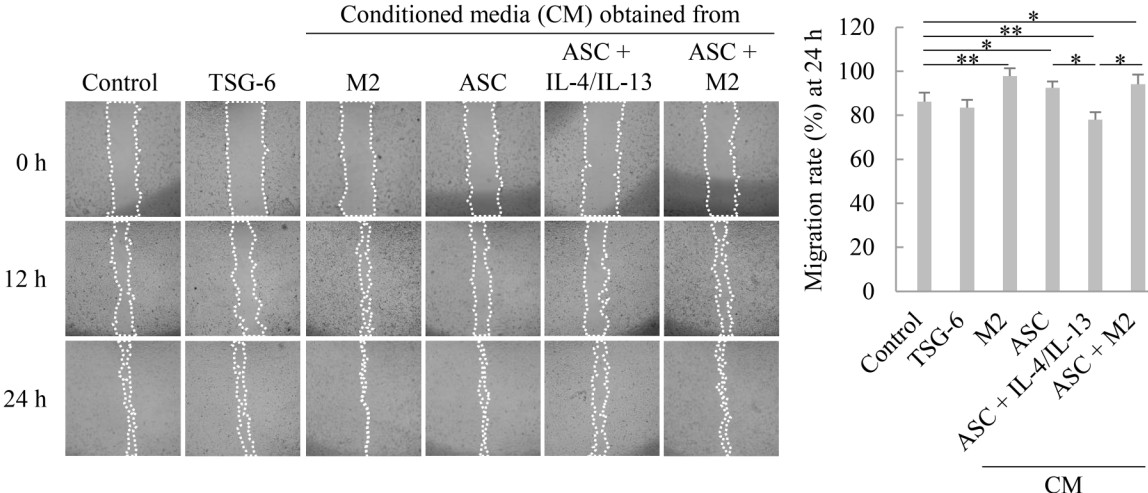

**Fig 4. Effects of adipose tissue-derived stem cells (ASCs) treated with interleukin (IL)-4/IL-13 or co-cultured with M2 macrophages on LX-2 cell migration.** Representative images from wound healing assay of LX-2 cells are shown from three independent experiments and the cell-free region (outlined wound area) was analyzed using ImageJ. Values of percentage wound closure were shown as the mean ± standard deviation (SD). *$p \leq 0.05$ and **$p \leq 0.01$. M2, M2 macrophage.

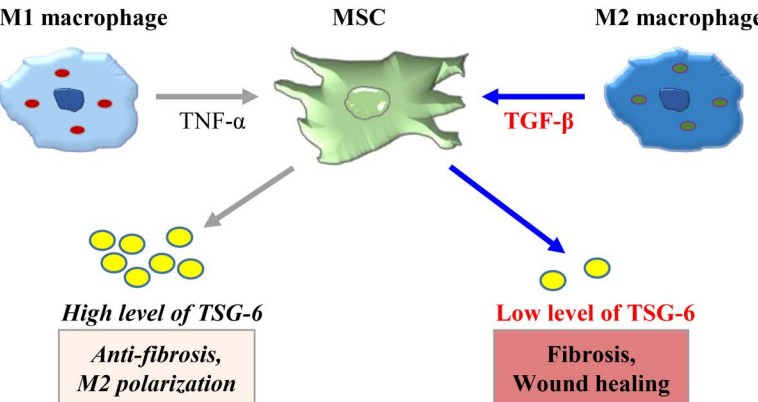

**Fig 5. Schematic representation of the therapeutic actions of mesenchymal stem cell (MSC) on fibrosis and wound healing.** Pro-inflammatory M1 macrophages induce overexpression of tumor necrosis factor-α (TNF-α) stimulated gene 6 (TSG-6) via TNF-α, which induces M2 polarization of macrophages and exerts anti-fibrotic actions. On the other hand, M2 macrophages can inhibit TSG-6 expression on MSCs via TGF-β, leading to fibrosis and wound healing.

excessive fibroblast proliferation and ECM accumulation Therefore, further studies are necessary to elucidate the precise mechanism of TSG-6 in wound healing.

MSCs have been used as cellular therapeutic agents in regenerative medicine because of their immunomodulatory and anti-fibrotic effects in various disease models. In wound healing, MSCs alleviate the inflammatory response in the early coagulation and inflammation phases of wound healing; regulate the proliferation of keratinocytes, fibroblasts, and host stem cells via hepatocyte growth factor (HGF), platelet-derived growth factor (PDGF), and vascular endothelial growth factor (VEGF) expression; and modulate the remodeling phase by regulating MMP and TIMP activities [37]. MSCs promote wound healing because TSG-6 is overexpressed by inflammatory M1 macrophages, and TSG-6 can convert macrophages

into M2 macrophages to alleviate LX-2 cell fibrosis [21,32]. However, excessive inhibition of the fibrotic activity of fibroblasts is not considered desirable from a wound healing perspective. From this perspective, inflammatory M1 macrophages induce MSCs to express TSG-6, which relieves inflammation and inhibits fibrosis during the inflammatory phase, but once the inflammation has resolved or M2 macrophages become active, TGF-β prevents MSCs from expressing TSG-6, allowing for fibroblast proliferation and ECM remodeling. However, MSC expression of TSG-6 alone is not sufficient to understand the mechanism of wound healing, as TGF-β and TNF-α expression can simultaneously be high in chronic fibrotic liver disease [38]. In other words, MSCs can promote fibroblast proliferation via growth factor expression such as VEGF, HGF, and PDGF, as well as promote wound healing by helping to regenerate new blood vessels.

In conclusion, M1 macrophages induce an anti-inflammatory response by promoting MSC TSG-6 expression in early wound healing, but as the wound progresses, M2 macrophages inhibit MSC TSG-6 expression to promote wound healing. However, a limitation of our study is that we only demonstrated in vitro that TSG-6 can delay normal wound healing. Further research is warranted to investigate the functions of TSG-6 during wound healing, especially during the different stages of wound healing in experimental animal models. In addition, step-wise transplantation of MSCs into different wound models and analyzing the TSG-6 expression of MSCs and phenotypic changes of macrophages is expected to help us understand the exact mechanisms of MSCs and macrophages in wound healing.

## Supporting information

**S1 Fig. M2 marker expression in macrophages treated with IL-4 and IL-13.** To differentiate M2 macrophages, PMA-treated THP-1 cells were treated with IL-4 (20 ng/mL) and IL-13 (20 ng/mL) for 48 h. Cells were then harvested using trypsin treatment and centrifugation. Macrophages were stained with fluorescein isothiocyanate (FITC)-conjugated anti-human CD163 antibodies or phycoerythrin (PE)-conjugated anti-human CD206 antibodies (both from BD Biosciences, San Jose, CA, USA) in the dark for 20 min at room temperature. FITC or PE-conjugated mouse immunoglobulin G was used as the isotype control at the same concentration. The fluorescence intensity of the cells was evaluated using flow cytometry (FACS Aria III Cell Sorter (BD Biosciences, Franklin Lakes, NJ, USA), and the data were analyzed using the FACSDiva Software v8.5 (BD Biosciences).
(TIF)

**S1 Raw images. Raw images of western blots and gels. Note: This file contains the original bands obtained in the western blot experiments generated by the study.**
(PDF)

**S1 Table. Tabular data.** Note: This document contains all the tabular data generated in the paper.
(XLSX)

## Author contributions

**Conceptualization:** Young Woo Eom, Pil Young Jung, Soon Koo Baik.

**Data curation:** Young Woo Eom, Pil Young Jung, Ki-Jong Rhee, Moon Young Kim, Soon Koo Baik, Hye Youn Kwon.

**Formal analysis:** Young Woo Eom, Ju-Eun Hong, Yongdae Yoon, Sang-Hyeon Yoo, Jiyun Hong.

**Funding acquisition:** Young Woo Eom, Pil Young Jung.

**Investigation:** Ju-Eun Hong, Yongdae Yoon, Sang-Hyeon Yoo, Jiyun Hong.

**Methodology:** Yongdae Yoon, Sang-Hyeon Yoo, Bhupendra Regmi, Saher Fatima.

**Project administration:** Moon Young Kim, Hye Youn Kwon.

**Supervision:** Hoon Ryu, Hye Youn Kwon.

**Validation:** Hye Youn Kwon.

**Visualization:** Young Woo Eom.

**Writing – original draft:** Young Woo Eom, Ki-Jong Rhee.

**Writing – review & editing:** Young Woo Eom, Ju-Eun Hong, Pil Young Jung, Sang-Hyeon Yoo, Jiyun Hong, Ki-Jong Rhee, Bhupendra Regmi, Saher Fatima, Moon Young Kim, Soon Koo Baik, Hoon Ryu, Hye Youn Kwon.

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
