## [Decision Letter · Decision Letter 0]

4 Feb 2025

PONE-D-24-56504TGF-β expressed by M2 macrophages promotes wound healing by inhibiting TSG-6 expression by mesenchymal stem cellsPLOS ONE

Dear Dr. Kwon,

Thank you for submitting your manuscript to PLOS ONE. After careful consideration, we feel that it has merit but does not fully meet PLOS ONE’s publication criteria as it currently stands. Therefore, we invite you to submit a revised version of the manuscript that addresses the points raised during the review process.

We look forward to receiving your revised manuscript.

Kind regards,

Zheng Yuan

Academic Editor

PLOS ONE

Journal Requirements:

“This research was funded by the Basic Science Research Program through the National Research Foundation of Korea (NRF) (RS-2023-00250982 and 2021R1I1A1A01056265).”

Reviewers' comments:

Reviewer's Responses to Questions

**Comments to the Author**

1. Is the manuscript technically sound, and do the data support the conclusions?

Reviewer #1: Partly

Reviewer #2: Yes

2. Has the statistical analysis been performed appropriately and rigorously? 

Reviewer #1: I Don't Know

Reviewer #2: Yes

3. Have the authors made all data underlying the findings in their manuscript fully available?

Reviewer #1: Yes

Reviewer #2: Yes

4. Is the manuscript presented in an intelligible fashion and written in standard English?

Reviewer #1: Yes

Reviewer #2: Yes

5. Review Comments to the Author

Reviewer #1: 1.This study lacks sufficient justification for selecting the TGF-β/TSG-6 signaling pathway as the primary focus. More detailed explanation is needed to clarify why this pathway is unique or important compared to other key factors like VEGF or PDGF.

2.The hypothesis section is not clear enough, especially regarding whether the role of TSG-6 in different stages of wound healing is consistent. It fails to distinguish between its functions during the inflammatory phase and the proliferative phase.

3.In the methods section, critical details about the co-culture system are missing, such as the pore size of the Transwell membrane and the exact composition of the culture medium. This lack of information may affect the reproducibility of the experiments.

4.The M2 macrophage phenotype validation is insufficient, relying solely on IL-4 and IL-13 induction. There is no flow cytometry or immunofluorescence data to confirm markers like CD206 and Arg1.

5.There is no indication that sample size was determined through power analysis, which may compromise the statistical reliability of the results. Further explanation of the statistical methodology is needed.

6.The presentation of figures is not very intuitive, especially for Figures 1 and 2. The figure legends lack sufficient detail, making it hard for readers to fully understand the experimental conditions and controls.

7.Western blot data lacks quantitative analysis, and no bar graphs summarizing relative protein expression levels are provided. This reduces the clarity and impact of the data presentation.

8.The study lacks in vivo experimental validation and relies solely on in vitro findings. This limits the translational potential and real-world relevance of the conclusions.

9.The mechanistic exploration is not deep enough. The study does not sufficiently investigate downstream signaling pathways, such as SMAD phosphorylation, which weakens the depth and completeness of the mechanistic explanation.

10.The discussion of clinical significance is too vague. It only briefly mentions therapeutic potential without offering specific application suggestions or feasible clinical strategies.

11.The language and formatting have issues, with some sentences being overly long or repetitive. For example, phrases like "where they infiltrate the wound, where they infiltrate the wound area" reduce the overall readability of the manuscript.

12.Some citations are not well-integrated into the context and do not clearly support the conclusions. These references need to be better linked to the narrative and their contributions clarified.

Reviewer #2: Overall, this paper presents clear results demonstrating the inverse correlation in expression levels of TSG-6 in MSCs and IGF-β in M2 macrophages, while also exploring the potential mechanisms involved in wound healing. It provides new insights into how the balance among TNF-α, TGF-β, and TSG-6 may play a crucial role in the wound healing process. However, there are several mistakes in the manuscript, I would like the authors to check and correct. First, Line 59, “where they infiltrate the wound, where they infiltrate the wound area.” This point is written twice here. Second, Line 246-247: However, TGF-β expression in M2 macrophages co-cultured with ASCs was lower than that in M1 macrophages cultured alone , but remained above the level expressed in macrophages. It should be: However, TGF-β expression in M2 macrophages co-cultured with ASCs was lower than that in M2 macrophages cultured alone , but remained above the level expressed in macrophages. My last point is question regarding to figure 2A, Based on this result that TGF- β 1 expression is decreased in co-cultured M2 compared to M2 alone, do you think that TGF- β 1 expressed is inhibited by ASC? If you treat M2 with TSG6, will the expression of TGF- β 1 go down?

6. PLOS authors have the option to publish the peer review history of their article (what does this mean? ). If published, this will include your full peer review and any attached files.

**Do you want your identity to be public for this peer review?** For information about this choice, including consent withdrawal, please see our Privacy Policy .

Reviewer #1: No

Reviewer #2: No

---

## [Author Response · Author response to Decision Letter 1]

25 Feb 2025

Thank you for providing clear and detailed comments and suggestions. They have proven immensely valuable in improving the quality of our manuscript. Building upon your input, we have expanded our analysis methods and made corresponding revisions to the discussion section. The introduction, methods, and conclusion were also revised. We believe that these revisions effectively address all the concerns raised. Please see the new version to assess the improvements made.

---

## [Decision Letter · Decision Letter 1]

10 Mar 2025

TGF-β expressed by M2 macrophages promotes wound healing by inhibiting TSG-6 expression by mesenchymal stem cells

PONE-D-24-56504R1

Dear Dr. Kwon,

We’re pleased to inform you that your manuscript has been judged scientifically suitable for publication and will be formally accepted for publication once it meets all outstanding technical requirements.

Kind regards,

Zheng Yuan

Academic Editor

PLOS ONE

Additional Editor Comments (optional):

Reviewers' comments:

Reviewer's Responses to Questions

**Comments to the Author**

1. If the authors have adequately addressed your comments raised in a previous round of review and you feel that this manuscript is now acceptable for publication, you may indicate that here to bypass the “Comments to the Author” section, enter your conflict of interest statement in the “Confidential to Editor” section, and submit your "Accept" recommendation.

Reviewer #1: All comments have been addressed

2. Is the manuscript technically sound, and do the data support the conclusions?

Reviewer #1: Yes

3. Has the statistical analysis been performed appropriately and rigorously? 

Reviewer #1: I Don't Know

4. Have the authors made all data underlying the findings in their manuscript fully available?

Reviewer #1: Yes

5. Is the manuscript presented in an intelligible fashion and written in standard English?

Reviewer #1: Yes

6. Review Comments to the Author

Reviewer #1: (No Response)

7. PLOS authors have the option to publish the peer review history of their article (what does this mean? ). If published, this will include your full peer review and any attached files.

**Do you want your identity to be public for this peer review?** For information about this choice, including consent withdrawal, please see our Privacy Policy .

Reviewer #1: No

---

## [Editor Report · Acceptance letter]

PONE-D-24-56504R1

PLOS ONE

Dear Dr. Kwon,

I'm pleased to inform you that your manuscript has been deemed suitable for publication in PLOS ONE. Congratulations! Your manuscript is now being handed over to our production team.

Kind regards,

on behalf of

Dr. Zheng Yuan

Academic Editor

PLOS ONE